# Diagnosis and Treatment Using Autologous Stem-Cell Transplantation in Primary Central Nervous System Lymphoma: A Systematic Review

**DOI:** 10.3390/cancers15020526

**Published:** 2023-01-15

**Authors:** Sara Steffanoni, Teresa Calimeri, Sarah Marktel, Rosamaria Nitti, Marco Foppoli, Andrés J. M. Ferreri

**Affiliations:** 1Department of Medicine, Division of Hematology, Valduce Hospital, 22100 Como, Italy; 2Lymphoma Unit, IRCCS San Raffaele Scientific Institute, 20132 Milan, Italy; 3Hematology and BMT Unit, IRCCS San Raffaele Scientific Institute, 20132 Milan, Italy

**Keywords:** autologous stem-cell transplantation, primary CNS lymphoma, consolidation therapy, neurotoxicity, conditioning regimen, busulfan, BCNU, thiotepa

## Abstract

**Simple Summary:**

Primary central nervous system (CNS) lymphoma (PCNSL), arising and remaining localized in the CNS, required the development of peculiar therapeutic strategies that deviate from those applied in systemic diffuse large B-cell lymphoma. To date, the optimal treatment approach for PCNSL consists in induction and consolidation/maintenance phases. Consolidation therapy with high-dose chemotherapy, followed by autologous stem-cell transplantation (HDC/ASCT), has demonstrated to be effective and safe in untreated and relapsed/refractory fit PCNSL patients; furthermore, it provides the preservation or improvement of cognitive function. This review offers scope to an overview of the experiences of HDC/ASCT as consolidation therapy in PCNSL patients, highlighting how conditioning regimens have changed over time. The progressive knowledge of CNS bio-availability of the single chemotherapy agents as well as of their efficacy and safety when used in different combinations has permitted to optimize the conditioning regimens with the unquestionable improvement of the outcome of the transplanted patients.

**Abstract:**

Background: Consolidation therapy has improved the outcome of newly diagnosed PCNSL patients. Whole-brain radiotherapy (WBRT) was the first consolidation strategy used and represented the gold standard for many years, but at the expense of a high risk of neurotoxicity. Thus, alternative strategies are being investigated in order to improve disease outcomes and to spare the neurocognitive side effects due to WBRT. Methods: We reviewed published studies on PCNSL patients treated with HDC/ASCT, focusing on the efficacy and safety of the conditioning regimens. Prospective and retrospective studies, published in the English language from 1992 to 2022, in high-quality international journals were identified in PubMed. Results: Consolidation with HDC containing highly CNS-penetrating agents (thiotepa, busulfan or BCNU) followed by ASCT provided long-term disease control and survival in PCNSL patients. Two prospective randomized studies, comparing HDC/ASCT versus WBRT, reported similar progression-free survival (PFS) and similar results on the decline in neurocognitive functions in a substantial proportion of patients after WBRT but not after HDC-ASCT. A recent randomized study comparing HDC/ASCT versus non-myeloablative consolidation reported a longer PFS in transplanted patients. Conclusion: ASCT conditioned with regimens, including highly CNS-penetrating agents, represents, to date, the best choice among the available consolidation strategies for fit newly diagnosed PCNSL patients.

## 1. Introduction

PCNSL is a rare variant of extra-nodal B-cell non-Hodgkin lymphoma, characterized by distinct biological features and clinical behaviors. PCNSL is highly chemo- and radio-sensitive. However, in order to reach an adequate cerebrospinal fluid (CSF) and CNS parenchyma concentration of the drugs, the use of agents in high doses and with a high CNS bio-availability has been the gold standard in PCNSL treatment. Currently, induction phase with HD-MTX-based chemotherapy regimens followed by consolidation/maintenance phase represents the standard first-line treatment for PCNSL. In addition, new drugs against PCNSL, such as those targeting the B-cell receptor signaling pathway, immunomodulatory drugs, checkpoint inhibitors and chimeric antigen receptor T cells (CART-cells), are undergoing clinical trials both in first and subsequent therapy lines. Despite the progressive improved outcome of PCNSL patients observed in recent decades, the prognosis of PCNSL remains unsatisfactory and is poorer than its counterpart represented by systemic diffuse large B-cell lymphoma (DLBCL). This review provides an overview of the efficacy and safety of HDC/ASCT in patients with PCNSL.

## 2. Role of ASCT in DLBCL and General Considerations on PCNSL Treatment

For many years, salvage platinum-based chemotherapy followed by HDC/ASCT has represented the standard of care in young and fit relapsed/refractory (R/R) patients affected by systemic DLBCL on the base of the results of the PARMA trial, which demonstrated a long-term disease-free survival in transplanted subjects [1]. Patients affected by DLBCL, who experienced a late relapse after first line (>12 months), had a better outcome after salvage therapy with HDC/ASCT compared to those who were refractory to first-line therapy or had an earlier relapse [2,3]. For DLBCL patients not responding to salvage chemotherapy or not eligible for HDC/ASCT, other approaches, such as CAR T-cell therapy and chemo-free therapies with antibody–drug conjugates (ADCs) as well as bispecific T-cell engagers (BiTEs), represent at present a promising alternative option to chemotherapy regimens.

In the last WHO classification of hematopoietic and lymphoid neoplasms, new separate lymphoma entities, including PCNSL, initially classified as DLBCL, have been recognized due to their peculiar clinical, biological and molecular features [4,5].

As in the case of R/R systemic DLBCLs, the first experiences on HDC/ASCT in PCNSL patients were applied in the R/R setting [6,7,8,9] (Appendix A). However, the poor general conditions of PCNSL patients at the moment of relapse and low response rate after salvage therapy limited the feasibility of HDC/ASCT in this setting. The results of a retrospective French real-life study on more than 1.000 immunocompetent R/R PCNSL patients showed that HDC/ASCT was performed only in 17% of the analyzed population, which included only chemo-sensitive, young (median age of 60 years) and fit patients [10]. Furthermore, another recent French real-life experience demonstrated that the number of ASCTs in PCNSL patients increased from 2016, especially in first-line treatment and in first-relapse, whilst it declined in subsequent lines, depicting the changing landscape of HDT/ASCT over time [11].

Based on the available evidence on the use of HD-MTX-based induction and HDC/ASCT as consolidation in the first-line setting [10,11,12,13,14], to date, age by itself cannot be considered as the only parameter in the treatment decision-making process for PCNSL. Instead of the “young” and “elderly” categories, we should select patients based on their ability to tolerate intensified treatments, considering performance status (PS), organ function, comorbidities and frailty. For fit patients eligible for HDT/ASCT, the most promising first-line therapy consists in high-dose MTX-based polychemotherapy (induction phase) followed by consolidation with HDC/ASCT, this option being less neurotoxic and almost as effective as WBRT [15,16,17]. On the other hand, for fit patients unsuitable for HDT/ASCT, the option of maintenance therapy with oral alkylating agent represents a valid alternative strategy to WBRT consolidation [18]. Primary WBRT, conferring a 2-year PFS of 30% [19], remains a therapy option only for unfit patients, for whom chemotherapy is contraindicated.

## 3. ASCT as Consolidation Therapy in Untreated PCNSL Patients

The aim of a consolidation therapy is to consolidate the gains obtained by induction therapy and to further eliminate any potential residual cancer cells, enhancing the likelihood of a durable complete remission and, thus, cure rate. In recent decades, for systemic DLBCL, consolidation with radiotherapy found indication for limited stages with PET-positive lesions after anthracycline-based regimens [20], while consolidation with HDC/ASCT was limited to R/R patients after salvage therapy [2,21].

Similar to systemic DLBCL, also for PCNSL, a consolidation therapy with WBRT became the standard of therapy in first-line treatment after induction chemotherapy. However, WBRT resulted to be complicated by a high rate of acute and delayed neurotoxicity, with a consequent worsening in quality of life despite a higher rate in disease control [19,22,23]. For this reason, after having demonstrated the efficacy and feasibility of HDC/ASCT in the R/R setting, this strategy was progressively investigated as consolidation in the first line as alternative to WBRT. The ability of hydrophilic chemotherapy agents to achieve a therapeutic dose in the CNS when used in high doses constituted the rationale of the applicability of HDC/ASCT in PCNSL as well. However, the use of combination regimens with non-cross resistant and highly CNS-penetrating agents demonstrated to be more promising. Furthermore, contrarily to what is observed with WBRT, growing evidence on neuro-cognitive preservation, and sometimes even improvement, is reported after HDC/ASCT [24,25]. Another advantage of HDC/ASCT over WBRT consists in the ability of the chemotherapy agents to penetrate and reach the entire CNS, including eyes, leptomeningeal space and spinal cord, conferring a consolidation in a “wider field” compared to WBRT.

### 3.1. Newly Diagnosed PCNSL Patients Eligible for HCD/ASCT

Most of the available evidence on the safety and efficacy of HDC/ASCT refers to young PCNSL patients (<60–65 years old) without severe comorbidities and without primary/acquired immunodeficiency. Recently, the German Group published the results of a multicenter pilot single-arm trial addressing the feasibility of HDC/ASCT in patients older than 65 years, with PS < 2, Cumulative Illness Rating Scale–Geriatric score < 6, no-active hepatitis and with normal organ function [26]. The consolidation therapy consisted of busulfan/thiotepa base-HDC (busulfan IV at 3.2 mg/kg on days -7 and -6 and thiotepa IV at 5 mg/kg on days -5 and -4) followed by ASCT. Only the cases in response or stable disease after 2 courses of Rituximab/HD-MTX/HD-cytarabine were transplanted. Thirty- seven patients aged over 65 years and with suspected newly diagnosed PCNSL were screened between December 2015 and September 2017. The screening process resulted in the inclusion of 14 patients, with a median age of 74 years (range of 69–79 years) and median PS of 1 (range of 0–2). All but one completed the therapy program, one patient stopped the treatment prematurely due to cardiac toxicity after the first cycle. A total of 4 patients achieved a complete response (CR)/ unconfirmed CR (CRu), 10 a partial response (PR) after induction chemotherapy. All 13 transplanted patients were in CR at 3 months after HDC/ASCT. With a median follow-up of 41 months, one patient in CR progressed at 9 months from transplantation. Two-year PFS and overall survival (OS) rates were 93% and 92%, respectively. The most frequent grade > 3 adverse events were infections and gastrointestinal disorders, and one serious adverse reaction after hematopoietic stem cells infusion was reported. All toxicities were reversible. No treatment-related death was observed.

HIV-positive patients have been excluded from key studies on the management of PCNSL. Data of the feasibility and efficacy of HDC/ASCT in the clinical setting of HIV-positive PCNSL patients are limited to a case report and a single center experience on five patients undergoing BCNU/thiotepa-conditioned ASCT after achieving a response with Rituximab/HD-MTX/HD-cytarabine-combined chemotherapy induction [27,28]. One patient developed Kaposi sarcoma progression, requiring liposomal doxorubicin therapy at +60 days after transplantation, and was in CR at +154 days. Two patients are in remission over 2 years post-transplant, a third patient is in CR at 7 months of follow up, one died for sepsis by Klebsiella pneumoniae and one died for Toxoplasma gondii encephalitis during aplasia [27]. Because of the limited data regarding the safety and efficacy of HDC/ASCT in HIV-related PCNSL patients, further evaluation and investigation are needed before drawing any conclusions and use in clinical practice.

It has been widely demonstrated that a higher survival rate and disease control were obtained with HDC/ASCT in B-cell lymphomas, achieving a response to previous induction or salvage chemotherapy [29,30,31]. For this reason, the study design of most trials and the recommendations of best clinical practice limit the consolidation with HDC/ASCT to PCNSL patients with responsive or stable disease after induction chemotherapy [32,33]. While an approach alternative to HDC/ASCT needs to be considered in chemo-refractory patients, the enrollment of these patients in clinical trials is strongly recommended.

#### 3.1.1. Upfront ASCT in PCNSL Single-Arm Studies

HDC/ASCT has been recently recognized as optimal consolidation approach alternative to WBRT in newly diagnosed PCNSL, sparing the WBRT related-neurotoxicity, which occurs in a range of 30–80% of cases, especially in patients older than 60 years [34,35]. The first evidence of safety and efficacy of HDC/ASCT performed as part of first-line therapy in PCNSL patients emerged in single-arm studies carried out from the early 2000s (Appendix A).

Different conditioning regimens are used in this setting of patients, which could be divided in:

(1) Melphalan-based regimens, similarly to those used for systemic DLBCLs. Because of being widely applied in systemic lymphoma and well known, these regimens were considered easily manageable. Because of low CNS penetration (of only 10% of the plasma concentration), in recent years, melphalan has been progressively abandoned as a conditioning agent for CNS lymphomas and replaced by more effective and CNS bio-available drugs [36].

(2) Busulfan-based regimens. Busulfan has a high CNS penetration achieving a CSF level of over 80% of plasma concentrations [36]. Furthermore, its high myeloablative and cytotoxic effect on lymphoma cells make it a promising drug in this setting.

(3) Alkylating agents-based regimens. BCNU and Thiotepa have an excellent CNS bio-availability (more than 80% of plasma level), high anti-neoplastic activity in malignant lymphoma and myeloablative effect. Furthermore, their non-cross-resistant efficacy has justified their use in combination.

The first experiences on HDC/ASCT in newly diagnosed immunocompetent PCNSL patients provided the use of BEAM (BCNU/etoposide/cytarabine/melphalan) regimen as conditioning schedule after induction therapy with HD MTX-based regimens and collection of *peripheral blood stem cells (PBSCs)* after *mobilizing* high-dose cytarabine-based chemotherapy [37]. The median age of the enrolled population was 51–53 years. Two studies provided a further consolidation with WBRT at a dose of 30 Gy with a boost of 10 Gy, in cases with residual disease after HDC/ASCT [38,39]. In all studies, about half of the population did not complete the treatment program with HDC/ASCT due to relapse/progression during induction therapy, particularly those treated with HD-MTX alone. The median number of *PBSCs* collected by leukapheresis was in the range of 25–32 × 10^6^ CD34+ cells/kg BW, and no marrow engraftment failure was reported. An absolute neutrophil count > 500/μL and platelet count > 20,000/μL were achieved after 8 and 9 days from *PBSCs* infusion, respectively. A low transplantation-related mortality was observed in all studies, and only one death before day +100 was reported by two studies [37,38]. The most frequent cause of death after ASCT was relapse/progression disease. All drugs used in BEAM conditioning regimen demonstrated to cross the blood–brain barrier, but the drug concentration in the brain appeared to be insufficient to induce an adequate cytotoxicity of malignant cells [37].

The LEED (melphalan/cyclophosphamide/etoposide/dexamethasone) conditioning regimen demonstrated its efficacy and safety in refractory systemic aggressive lymphoma [40,41]. The results on the efficacy and safety of LEED conditioned ASCT in PCNSL are limited to a retrospective study on 13 patients treated between April 2004 and June 2013. The study provided an induction therapy with three cycles of MPV (HD-MTX/ procarbazine/vincristine) plus two cycles of high-dose cytarabine followed by *PBSCs* harvest. In the case of residual disease after ASCT, WBRT or local irradiation was considered. All 13 patients achieved a response after induction therapy. Three patients did not complete the therapeutic program with ASCT due to *PBSCs* collection failure, and four interrupted the conditioning chemotherapy due to seizure (*n* = 2) or viral infection (*n* = 2). Only 6 out 13 patients were transplanted. Considering the subgroup of transplanted patients, 3-year OS rate was 80% and the 3-year PFS rate was 83%. All patients achieved stem-cell engraftment, with a median time to neutrophil recovery of 13 days (range of 10–22 days). Two patients experienced grade 3 febrile neutropenia after ASCT, all patients had low-moderate hepatic dysfunction and two patients had grade 2 oral mucositis. All ASCT-related toxicities were reversible. No delayed adverse events, including neurotoxicity, were reported. The limited number of patients precluded final considerations on the efficacy and tolerability of LEED conditioning chemotherapy in PCNSL [42].

In PCNSL patients, melphalan-based-conditioning regimens were progressively replaced by more efficient schedules containing agents with higher CNS penetrance.

BuCyE (Busulfan/cyclophosphamide/etoposide), commonly used as conditioning regimen before allogeneic hematopoietic stem-cell transplantation (allo-SCT), has been used in ASCT setting in systemic lymphoma, conferring a long-term survival rate of 43–58% [43,44]. An observational single-center study compared the toxicity profile and efficacy of BEAM versus BuCyE conditioning regimens in lymphoma patients (including 11 Hodgkin lymphoma and 31 non-Hodgkin lymphoma). No significant differences in toxicity and efficacy (OS and EFS) between two regimens were observed; however, a trend to a higher incidence of mucositis in the BuCyE subgroup was reported [45].

The use of BuCyE regimen as conditioning chemotherapy followed by ASCT rescue in newly diagnosed PCNSL is limited to a single center experience on 11 patients. All patients were transplanted in response (10 CR/uCR and 1 PR) after induction therapy with five courses of HD-MTX and two cycles of HD-cytarabine. Two patients received additional WBRT after HDC/ASCT. A total of 72% of the cases (8/11) experienced a transferase elevation after conditioning chemotherapy and 18% (2/11) low-moderate hyper-bilirubinemia. Two patients (18%) experienced grade 1/2 mucositis and two (18%) grade 3 diarrhea. Febrile neutropenia was reported in 10 (91%) patients. No case of veno-occlusive disease was observed. Engraftment of neutrophils and platelets was achieved at a median of 9 days (range of 8–12) and 7 days (range of 5–12), respectively. After a median follow-up of 25 months, six patients had relapsed (54%), with a median event-free interval of 15 months and 2-year OS rate of 89% [46]. Considering its high rate of relapse, BuCyE should not be considered as a standard conditioning regimen before ASCT in PCNSL patients.

A higher response rate with prolonged CR was obtained by using the TBC (thiotepa/busulfan/cyclophosphamide) conditioning regimen. TBC schedule, however, was graved by a high incidence of treatment-related complications, particularly in patients older than 60 years. For this reason, its use should be restricted to experienced centers and an accurate patient selection is highly recommended. The larger single-arm study investigating the safety and efficacy of TBC conditioning regimen included 33 newly diagnosed PCNSL patients, who were candidate to ASCT if in response after induction therapy with 5–7 cycles of R-MPV (rituximab/HD-MTX /procarbazine/vincristine) [47]. A total of 26 out of 33 patients were transplanted. Two-year PFS and OS were both 81% in the transplanted subgroup. A significant improvement in cognitive functions and performance status was observed in most of the patients, and no neurologic decline was recorded after ASCT. Anti-epileptic prophylaxis and the use of mesna to prevent cyclophosphamide-related hemorrhagic cystitis are highly recommended. In order to reduce the incidence of hepatic and infection toxicity, a supportive care with ursodiol, antiviral (acyclovir or famciclovir), anti-pneumocystis and anti-fungal prophylaxis was performed in most of the studies involving TBC chemotherapy use [48,49,50]. Neutrophil and platelet engraftment was reached after a median time of 9–11 days. The median duration of hospitalization for ASCT was in the range of 23–28 days. Regarding the toxicity related to TBC regimens, grade 3/4 mucositis was the most frequent complication, occurring in up to 81% of the cases. Grade 3 febrile neutropenia and bacterial or fungal infections were recorded in about 40% and 20% of patients, respectively. No cases of significant long-term cognitive dysfunction or decline were reported. No-relapse mortality (NRM) and therapy-related mortality (TRM) resulted to be up to 24% and 19%, respectively. Factors associated with NRM included age over 60 years, pre-existing immune suppression, and the era in which transplantation was performed (1998–2003 versus 2000–2010).

An effort of improvement upon the efficacy of TBC conditioning regimen was attempted by adding high-dose intra-venous rituximab (1 gr/m^2^) administered on day −9 (after starting thiotepa) and day −2 (after the completion of cyclophosphamide). A total of 30 CNS lymphoma patients (18 PCNSL and 12 SCNSL) were enrolled prospectively to receive consolidation therapy with HDC/ASCT after HD-MTX-based chemotherapy [51]. The most common toxicities observed during aplasia were diarrhea and mucositis. All patients experienced febrile neutropenia, bacterial infection was documented in 9/30 cases, and 1 case had severe candidemia and cytomegalovirus reactivation. Two patients developed significant neurotoxicity, and one of them was previously treated with WBRT. Considering the subgroup with PCNSL (*n* = 18, including 6 R/R PCNSL), the safety and efficacy of this approach were demonstrated, and no death or relapse was reported after a median follow-up of 24 months from ASCT. However, a longer follow-up is needed before the use of TBC plus HD Rituximab conditioning schedule in clinical practice.

A less toxic regimen without cyclophosphamide as conditioning chemotherapy (Busulfan/thiotepa: Busulfan 16 mg/kg BW and thiotepa 10 mg/kg BW) was investigated in 16 newly PCNSL patients, after having received two cycles of HD-MTX (8 g/m^2^) as induction. Nine of the patients received further consolidation with WBRT after HDC/ASCT. Two-year OS and EFS were 61% and 48%, respectively. Two patients died of infections. A median of 4.1 × 10^6^ CD34+ cells/kg BW (range of 2–9) were reinfused with a median time to leucocyte recovery (>1000/mmc) of 8 days (range of 7–12) and to thrombocyte recovery (>50,000/mmc) of 11 days (range of 8–15) [52].

The feasibility and efficacy of BCNU/thiotepa (BCNU 400 mg/m2 on day -6 and thiotepa 5 mg/kg on days -5 and -4) as a conditioning regimen before ASCT were firstly investigated in 30 untreated PCNSL young patients (<65 years old) [53]. The design of the study provided a sequential induction chemotherapy with three cycles of HD-MTX/HD-cytarabine/thiotepa followed by a combined consolidation strategy with HDC-ASCT and hyper-fractionated WBRT (hWBRT) at a dose of 45 Gy in CR patients or 50 Gy in PR patients. A total of 33 out of 30 patients proceeded to HDC/ASCT and 21 out 23 patients to subsequent hWBRT. The median time of leucocyte recovery was 7.5 days (range of 5–11). Thrombocyte recovery (>20,000/mmc) occurred in 19 of 23 patients with a median duration of thrombocytopenia of 1 day (range of 0–8). Febrile neutropenia occurred in 12 out 23 patients, and 1 patient experienced suspicious fungal pulmonary infiltrates and six patients grade < 2 mucositis. No TRM was reported. After a median follow-up of 63 months (range of 4–84 months), 5-year OS was 69% for the entire population and 87% for the transplanted patients. The five-year relapse-related death rate was 21% for the entire population and 9% for patients receiving HDC/ASCT. Five patients developed leukoencephalopathy, and in all cases, this was attributed to hWBRT. The same authors later published the results of prospective study on 43 immunocompetent, young (<67 years), newly diagnosed PCNSL patients, treated from 1998 to 2006 with 3–4 courses of HD-MTX (at dose of 8 g/m^2^) followed by one or two cycles of high-dose cytarabine/thiotepa and subsequent *PBSCs mobilization*. The consolidation phase consisted in BCNU/thiotepa (thiotepa 5 mg/kg for two doses in 30 patients and 5 mg/kg for four doses in 10 patients)-conditioned ASCT ± WBRT. WBRT was restricted to subjects who did not achieve CR after ASCT. Thirty-four patients (79%) achieved a CR after first-line treatment. Twelve (35%) relapsed, half of them after 5 years from diagnosis. Only 10 patients were not irradiated and all of them achieved a response (nine CR and one PR) [54]. Considering the entire enrolled population, OS and EFS at 2 years were both 81%, while at 5 years, EFS and OS were 70% and 67%, respectively. No TRM was reported.

Sequential HD-MTX-based chemotherapy followed by BCNU/thiotepa-containing HCT-ASCT demonstrated to be a promising treatment option, leading to remarkable median survival rates and a favorable toxicity profile.

The results of a retrospective monocenter study on 247 patients affected by aggressive B-cell lymphomas (including mantle cell lymphoma, systemic DLBCL, Hodgkin lymphoma and PCNSL) receiving HDC/ASCT between 2002 and 2019 were recently reported. The population included 45 patients with PCNSL (35 newly diagnosed and 10 R/R) [29]. All patients with PCNSL received BCNU/thiotepa as the conditioning regimen. The subgroup of PCNSL patients consolidated with HDC/ASCT in first line had a significantly improved PFS and OS compared to those transplanted in the second or further lines. This finding supports the actual evidence of the use of HDC/ASCT as part of the first-line treatment in PCNSL with the aim to reach a cure rate or at least long-term disease control in about 60–75% of the patients. Infections and mucositis represented the most relevant adverse events after ASCT, occurring in 40% and 80% of the cases. In a minority of cases (2%) the infections resulted to be fatal. Death after ASCT was mainly related to relapse disease, while day +100 NRM was less than 5%. Considering the entire cohort, the Authors reported a higher NRM rate in patients receiving lower numbers of transfused stem cells with a potentially higher risk for infectious complications, despite no difference in time to leukocyte engraftment being recorded according to the number of transfused stem cells. However, the number of infused stem cells might be a surrogate parameter for impaired immune cell recovery with a consequent higher susceptibility to severe infections [29].

As reported below, the first experience of HDC/ASCT included BEAM schedule as the conditioning regimen with poor results both in terms of EFS (43–46%) and OS (60%) at 2 years and with a high incidence of mortality secondary to disease progression (more than 50%) [37,38]. More recently, regimens with highly CNS-penetrating agents, such as BCNU, thiotepa and busulfan, have been used with more encouraging results, achieving a 2-year PFS and OS of about 80% after transplantation [53,54]. Although there are no studies directly comparing different conditioning regimens (BEAM versus BCNU/thiotepa versus thiotepa/busulfan versus thiotepa/busulfan/cyclophosphamide), the overall response rates and PFS appeared to be better with thiotepa-based conditioning regimens. By multivariate analysis, TBC regimen conferred a better outcome than other regimens but at the cost of a higher therapy-related mortality [55]. To date, there is not a standard conditioning regimen that is widely approved for PCNSL patients. However, the host characteristics, the previous chemotherapy performed and their outcome as well as the HDC/ASCT-related potential toxicity should be considered at the moment of choosing the conditioning regimen.

Early relapse, persistent disease before ASCT and the older age demonstrated to be predictive factors of a poorer outcome in transplanted systemic DLBCL patients [2,30]. Regarding PCNSL patients, to date, no data on risk factors predicting post-ASCT outcomes have been validated and widely adopted.

#### 3.1.2. Upfront HDC/ASCT in PCNSL: Randomized Studies

In the last two decades, researchers have focused their efforts to establish the best consolidation strategy in PCNSL patients that is able to confer a high rate of disease control and longer survival with a limited risk of acute and delayed toxicity. IELSG32 and PRECIS randomized prospective trials compared two strategies (HDC/ASCT and WBRT) in young immunocompetent patients after HD-MTX-based induction chemotherapy [16,17]. In the IELSG32 trial, patients (<70 years old) with responsive/stable disease after the induction phase were randomized between WBRT (*n* = 59) and BCNU/thiotepa (5 mg/kg BID days -5 and -4) conditioned ASCT (*n* = 59). Consolidation with WBRT and HDC/ASCT exhibited a similar efficacy in terms of both PFS (55% for WBRT vs 50% for HDC/ASCT) and OS (63% vs. 57%), after a median follow-up of 7 years [56]. As expected, hematological toxicity was more common in the transplanted subgroup, and grade 4 non-hematological toxicity (coagulopathy, infection, hepatotoxicity and mucositis) occurred in about 5% of patients of both arms. Acute neurotoxicity was grade 3 or less, and more commonly reported among irradiated patients (18% vs. 7%, *p* = 0.089). Two toxic deaths due to infection occurred within +100 days after ASCT. Among the patients free from relapse, 14 deaths were reported: 4 for infections, 4 for unknown cause, 3 for cognitive decline, 2 for second tumor and 1 for car accident. Second malignancies occurred in eight patients (4%): five in WBRT-arm patients and three in HDC/ASCT-arm. Of note, deaths in the relapse-free patients and the incidence of second tumors were not significantly related to consolidation strategies. Neuropsychological tests showed a statistically significant impairment in some attentive and executive functions in irradiated patient, while a significant functional/memory and quality-of-life (QoL) improvement was observed in the transplanted patients [56]. Based on the IELSG32 results, both WBRT and HDC/ASCT should be considered feasible, safe and effective consolidation strategies after HD-MTX-based induction in newly diagnosed PCNSL. However, ASCT should be preferred given the neurological impairment observed after WBRT. Patients with meningeal involvement had a trend to improved PFS in favor of consolidation with ASCT compared to WBRT (67% versus 40% at 7-year), a difference that did not reach significant levels (*p* = 0.32). For this subgroup of patients, the use of WBRT as a consolidation strategy should be used with extreme caution.

The study design of the PRECIS trial provided the enrollment of young patients (18–60 years old) with newly diagnosed PCNSL and the randomization (1:1) at the time of study registration to receive a consolidation treatment either with WBRT (40 Gy; 2 Gy/fraction) or with TBC-conditioned ASCT (arm B). Induction phase included two cycles of R-MBVP (rituximab/HD-MTX/BCNU/etoposide/prednisone) and two cycles of Rituximab/HD-cytarabine. Both consolidation treatments achieved the predetermined efficacy threshold, but a better 2-year PFS was achieved after HDC/ASCT (87% vs. 69%) [16]. This trend was confirmed by long-term data (after a median follow-up of 8 years), which showed a lower EFS from randomization in the WBRT arm with respect to the HDC/ASCT arm (39% vs. 67%, *p* = 0.03) and a statistically significant lower risk of relapse after HDC/ASCT (hazard ratio = 0.13, *p* < 0.001) [57]. Febrile neutropenia occurred in all patients during aplasia. No oral or gastro-intestinal mucositis of grade >3 was reported. Electrolyte/glycemic/hepatic disorders occurred in more than 60% of the cases and neuropsychiatric disturbances in 16% of the entire population. No significant difference in terms of 8-year OS was observed between two groups (69% HDC/ASCT versus 65% WBRT), because one-third of patients who relapsed after WBRT were still alive after a salvage treatment. A higher incidence of neurocognitive deterioration was reported after WBRT consolidation compared to ASCT during the follow-up (64% versus 13%, *p* < 0.001). On the basis of these data, the authors concluded that 40 Gy WBRT should not be recommended as a consolidation therapy in first-line treatment for PCNSL because of highly neurotoxic and suboptimal in reducing the risk of relapse, while HDC/ASCT appears to be highly efficient in preventing relapses.

Non-myeloablative chemotherapy was investigated as alternative consolidation strategy to HDC/ASCT. The schedule consisted in agents with high CNS bio-availability and with a mechanism of action different from MTX. In the prospective CALGB 50202 trial and retrospective study, a combined regimen with etoposide and HD-cytarabine (EA schedule) was used as no-myeloablative consolidation strategy in patients with newly diagnosed PCNSL and who were in response after induction therapy. Promising results with 2-year time to progression (TTP) of 69% and 2-year PFS of 83% were reported [58,59]. Based on these results, two randomized trials comparing the efficacy of no-irradiation consolidative strategies (non-myeloablative chemotherapy versus HDC/ASCT) were carried out by the IELSG and CALGB groups. The randomized phase II CALGB (Alliance) 51101 trial enrolled young and fit PCNSL patients (median age of 61 years; range of 33–75), who were randomized at the time of study registration to receive BCNU/thiotepa-conditioned ASCT (arm A) versus EA chemotherapy (arm B). A total of 70 out of the 113 enrolled patients completed the consolidation per protocol. After a median follow-up of 3.8 years, the median PFS from randomization was 6 years in arm A and 2.4 years in arm B. Of note, in the non-myeloablative chemotherapy arm, a major rate of events (such as progression and death) occurred before the consolidation therapy and this introduces inevitable bias in the results. The toxicities were similar between two arms, and no TRM during the consolidation was observed [60]. The IELSG43 multicenter randomized trial was designed to compare BCNU/thiotepa-conditioned ASCT versus non-myeloablative chemotherapy with DeVIC (dexamethasone/etoposide/ifosfamide/carboplatin) schedule plus rituximab. The enrolled patients (aged < 70 years) were randomized to consolidation chemotherapy only after having achieved at least stable disease after four courses of MATRix (HD-MTX/HD cytarabine/thiotepa/Rituximab) (NCT02531841). The results were recently presented. After a median follow-up of 44 months (range of 0,2–86), the consolidation with HDC/ASCT resulted in a significantly better outcome than non-myeloablative chemoimmunotherapy with 3-year PFS of 79% versus 53% (HR 0.42; *p* = 0.0003). The 3-year OS was 86% for the HDC-ASCT arm and 71% for the R-DeVIC arm (HR 0.47; *p* = 0.01). No significant difference in neurocognitive function evaluation was registered between two arms [61].

The advantages of a consolidation with non-myeloablative chemotherapy over HDC/ASCT consist in (1) reduction in the side effects secondary to prolonged pancytopenia, such as severe infections and risk of bleeding as well as the reduction in blood transfusion support, and (2) a less resource-consuming strategy and a shorter duration of hospitalization. However, in particular, in view of the results of the IELSG 43 trial, HDC/ASCT should be considered the standard consolidation therapy for fit PCNSL patients [61]. To date, consolidation with non-myeloablative chemotherapy might find applicability in clinical practice only in selected cases considered fit for high-dose chemotherapy but not eligible for transplantation due to causes other than patient fitness (e.g., due to poor mobilization).

## 4. HDC/ASCT and Neurotoxicity

Therapy-related neurocognitive impairment consists in negative changes in neurological function that are independent of normal aging and affect the activities of daily living and QoL. Clinical and neuroimaging manifestations suggestive of neurotoxicity are observed in long-term PCNSL survivors after WBRT and, in recent decades, it has become a major concern due to a longer survival of PCNSL patients.

WBRT-associated delayed neurotoxicity especially affects elderly patients. In recent years, the investigation of alternative, effective, no irradiation consolidation (HDC/ASCT or non-myeloablative chemotherapy) and maintenance strategy (such as temozolomide) have been the objective of studies of several research groups. The assessment of cognitive function was provided both in IELSG 32 and PRECIS trials and was performed by standard instruments at different timepoints (at diagnosis, at the end of first-line therapy and during follow-up). The final results demonstrated a significant advantage in neurocognitive improvement in transplanted patients with respect to those receiving WBRT as consolidation therapy in first line [24,56,57].

A less recent prospective study investigated cognitive function among PCNSL patients in long-term remission after first-line consolidation therapy with a reduced dose (rd) WBRT or with HCT/ASCT. The neurological tests were performed at diagnosis, post-induction and yearly after consolidation up to 5 years [24]. A significant improvement in several cognitive domains was observed from baseline up to 3 years. However, a later decline in attention/executive functions and memory was reported in both subgroups with delayed abnormal features in brain MRI (such as the appearance of diffuse alteration of white matter, ventricular dilatation and cortical/subcortical atrophy).

Change and/or decline in neurocognitive functions resulting from impaired hippocampal neurogenesis might occur after WBRT. It was reported that conformal hippocampal sparing could provide the preservation of neurocognitive functions. Modern Volumetric Modulated Arc Therapy techniques, able to reduce the dose irradiating bilateral hippocampi below the dosimetric threshold, was applied in patients undergoing WBRT for the CNS metastasis by solid tumor or for primary glioma tumor with promising results [62,63]. Before its applicability in patients with PCNSL, patients should be recruited in prospective trials of hippocampal sparing during cranial irradiation to accomplish neurocognitive preservation while maintaining intracranial control. However, to be considered that deep cerebral areas, including the hippocampus, are often involved by PCNSL, the application of this radiation strategy could result in under-dosing the tumor target.

The neurologic damage due to lymphoma CNS infiltration plays a crucial role in the final neurocognitive assessment. For this reason, the development of new diagnostic approaches is the object of research in order to limit brain damage due to disease and to the diagnostic biopsy. For this reason, it is extremally important to investigate more accurate, less invasive and worldwide-performed diagnostic approaches that can be an alternative to the actual standard diagnostic options (such as stereotactic biopsy, detection of molecules or/and lymphoma cells on CSF) with the aim to provide earlier diagnosis with a minimally invasive procedure.

## 5. HDC/ASCT and PCNSL Patients: Open Questions

Although there are many improvements that have been achieved in the management of PCNSL, reached in recent decades with a consequent progressively longer survival, some questions remain unanswered and could be the objectives of future clinical trials.

(1) Biological age did not demonstrate to be a reliable and unquestionable criterion for distinguishing patients eligible for HDC/ASCT or not. For this reason, a score, including host characteristics and prognostic factors, could be a useful tool to discriminate patients older than 70 years who could benefit from consolidation with HDC/ASCT.

(2) Unlike what happens in some lymphoproliferative disorders such as Hodgkin lymphoma and mantle cell lymphoma, where the maintenance therapy after HDC/ASCT became a standard of care in some clinical conditions, for PCNSL patients, the role of maintenance therapy after HDC/ASCT still remains to be investigated.

(3) The optimal thresholds of autograft cellular composition to be infused for PCNSL needed to be better established with prospective and retrospective studies. A recent prospective multicenter study on 17 PCNSL patients receiving HDC/ASCT upfront, focusing on the effects of blood graft cellular content on hematologic recovery and outcome, was published [64]. Most of the patients received a combined systemic and intraventricular chemotherapy regimen according to the Bonn protocol (*n* = 11), and the remaining patients were treated with MATRix chemotherapy (*n* = 6) as the induction therapy. The results showed that a higher number of infused viable (CD34+ cells > 1.7 × 10^6^/kg) was linked to a more rapid platelet engraftment (10 vs. 31 days, *p* = 0.027) and neutrophil recovery at day +15 (5.4 vs. 1.6 × 10^9^/L, *p* = 0.047). A higher amount of cytotoxic T CD3+ cells infused had an impact on both the hematologic recovery with slower neutrophil engraftment and a lower number of platelet recovery both at day +15 and at 1 month after the graft infusion. A higher number of total graft CD34+ yield collected with a cut-off point of >3.3 × 10^6^/kg was associated with a worse 5-year PFS (33% vs. 100%, *p* = 0.028), but the number of viable CD34+ cells after thawing with a cut-off point of > 1.7 × 10^6^/kg did not correlate with PFS (*p* = 0.688) or OS (*p* = 0.414). In addition, >78 × 10^6^/kg CD3 + CD8+ T cells in the infused graft impacted negatively on the 5-year PFS (0% vs. 88%, *p* = 0.016). However, these results should be handled with caution due to small number of patients.

(4) The best management of patients not achieving a complete response after transplantation remains to be defined. In some single-arm studies, further consolidation with WBRT was restricted to patients without CR after HDC/ASCT because of the risk of RT-related neurotoxicity. In the future, the use of novel agents, such as target therapy, immunomodulatory drugs and/or PD1 inhibitors, should be investigated in this setting of patients as a maintenance or consolidation strategy with the aim of reaching a deeper and durable response.

## 6. Conclusions

There are three available consolidation strategies that can be applied in PCNSL as part of the first line of therapy. They are WBRT, non-myeloablative chemotherapy and HDC followed by ASCT rescue. There are two main advantages of HDC/ASCT over WBRT: (1) avoidance of radiation-induced neurotoxicity and (2) the ability of the chemotherapy agents to penetrate and reach the entire CNS, including eyes, leptomeningeal space and spinal cord, conferring a consolidation in a “wider field” than what can be achieved with WBRT. Additionally, WBRT has some advantages over chemotherapy that should be taken into consideration when choosing the most appropriate therapeutic strategy. It provides a lower risk of hematological toxicity and, consequently, of infective complications as well as limited liver and renal toxicities. Furthermore, WBRT does not require hospitalization. The advantages of consolidation with non-myeloablative chemotherapy over HDC/ASCT consists in (1) the reduction in the side effects secondary to prolonged pancytopenia, such as severe infections and risk of bleeding as well as the reduction in blood transfusion support, and (2) a less resource consuming strategy and a shorter duration of hospitalization. To date, based on the results of randomized studies [16,17,60,61], HDC with regimens including highly CNS-penetrating agents followed by ASCT rescue should be considered the standard consolidation strategy for fit PCNSL patients in response or stable disease after induction therapy, due to its promising outcome without any measurable negative effect on neurocognitive functions and with an excellent risk-to-benefit ratio.

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
