# Peer review of "Diagnosis and Treatment Using Autologous Stem-Cell Transplantation in Primary Central Nervous System Lymphoma: A Systematic Review"

_cancers, 2023, doi:10.3390/cancers15020526_

Round 1

Reviewer 1 Report

Presented review is an interesting and well-written one. Moreover its potential clinical implication may be of value for PCNSL patients. This analysis sheds light on the recurrence rate in lymphoma patients treated between 1992 and 2022.

Please address all comments: 

- On page8, first paragraph: ... acute neurotoxicity was grade 3 or less, and more commonly reported among irradiated patients (18% vs 7%). Was this diffenrence significant? please report on P- value.

- Authors should include less radiotherapy dose-deescalation (as an alternative and modern radiotherapy option with less neurotoxicity) in the discussion section. Potential radiotherapy options with less toxicities include: local radiotherapy or whole-brain irradiation with hippocampal sparing.

- One important point is the financial toxicity. The authors should compare between the possible cost difference between HCT/ASCT and WBRT, which is very important in many patients all over the world.

- Table 1 and 2: please add name of first author and the year of publication to first column (for example, Soussain et al. 2001) to be more clear for readers.

- The authors added the dose of administered chemotherapy in table 1 and 2, however the WBRT dose not provided in the tables. Please add radiotherapy dose applied in study 38,47, 53, 16, 17.

Author Response

Dear Reviewer,  please following can you find our replies to all your comments/suggestions:

- On page8, first paragraph: ... acute neurotoxicity was grade 3 or less, and more commonly reported among irradiated patients (18% vs 7%). Was this difference significant? please report on P- value.

Reply: The value has been reported as suggested

- Authors should include less radiotherapy dose-deescalation (as an alternative and modern radiotherapy option with less neurotoxicity) in the discussion section. Potential radiotherapy options with less toxicities include: local radiotherapy or whole-brain irradiation with hippocampal sparing.

This topic has been added and discussed in the neurotoxicity section (please see page 9)

- One important point is the financial toxicity. The authors should compare between the possible cost difference between HCT/ASCT and WBRT, which is very important in many patients all over the world.

This issue has been added and discussed in the conclusions (please see page 11)

- Table 1 and 2: please add name of first author and the year of publication to first column (for example, Soussain et al. 2001) to be more clear for readers.

This additions was made in table 1 and 2, as suggested

- The authors added the dose of administered chemotherapy in table 1 and 2, however the WBRT dose not provided in the tables. Please add radiotherapy dose applied in study 38,47, 53, 16, 17.

The radiotherapy doses applied have been reported in the tables.

Sincerely

Reviewer 2 Report

This review on the role of high-dose chemotherapy and autologous stem cell transplantation in primary CNS lymphoma gives a comprehensive overview of the available literature. However, there are few minor points that should be addressed by the authors:

1) what about the results of the IELSG43 trials presented at ASH 2022? These should be included and discussed.

2) the references are not indicated in some paragraphes. Please revise.

Author Response

Dear Reviewer,  please following can you find our replies to all your comments/suggestions:

1) what about the results of the IELSG43 trials presented at ASH 2022? These should be included and discussed.

The results of IELSG 43, recently presented, were reported and discussed in the paragraph 3.1.2 Upfront HDC/ASCT in PCNSL: randomized studies

2) the references are not indicated in some paragraphs. Please revise.

It has been revised and corrected

Sincerely